# Insights into Mobile Small-RNAs Mediated Signaling in Plants

**DOI:** 10.3390/plants11223155

**Published:** 2022-11-18

**Authors:** Yan Yan

**Affiliations:** Department of Biochemistry and Biophysics, Texas A&M University, College Station, TX 77843, USA; yanyan-2019@tamu.edu

**Keywords:** RNAi, small-RNA, miRNA movement

## Abstract

In higher plants, small RNA (sRNA)-mediated RNA interfering (RNAi) is involved in a broad range of biological processes. Growing evidence supports the model that sRNAs are mobile signaling agents that move intercellularly, systemically and cross-species. Recently, considerable progress has been made in terms of characterization of the mobile sRNAs population and their function. In this review, recent progress in identification of new mobile sRNAs is assessed. Here, critical questions related to the function of these mobile sRNAs in coordinating developmental, physiological and defense-related processes is discussed. The forms of mobile sRNAs and the underlying mechanisms mediating sRNA trafficking are discussed next. A concerted effort has been made to integrate these new findings into a comprehensive overview of mobile sRNAs signaling in plants. Finally, potential important areas for both basic science and potential applications are highlighted for future research.

## 1. Introduction

Mobile small RNAs (sRNAs) are sequence-specific signaling agents regulating plant responses to diverse internal developmental and external environmental cues. These are three major groups of sRNAs in plants, small interference RNA (siRNAs), microRNAs (miRNA), and transfer RNA-derived fragments (tRFs), which are 21 to 24 nucleotides (nt) in size [1,2,3,4]. The siRNAs are generated through processing double-stranded RNAs (dsRNAs), either inverted-repeat (IR) transcripts or RNA-dependent RNA polymerases (RDRs) synthesized dsRNAs, by Dicer-like2 (DCL2), DCL3 or DCL4 [5]. miRNAs are generated by DCL1 cleavage of imperfectly paired harpin of native transcript precursors [6]. A type of newly identified sRNAs are derived from RNases T2 catalyzed precursor or mature transfer RNAs (tRNA). These tRFs are present in diverse organisms at a comparable level to miRNAs [7,8]. These sRNAs can be integrated into Argonaute proteins (AGOs), forming the core of RNA-induced silencing complexes (RISC), to mediate either transcriptional gene silencing (TGS) or posttranscriptional gene silencing (PTGS) [7,9,10,11].

These non-cell-autonomous sRNAs were reported moving cell-to-cell through plasmodesmata (PD), bidirectional systemically via vasculature to distal organs and parasites, or cross-kingdoms by extracellular vehicles (EVs) to pathogens [2,12,13,14,15,16,17,18]. Here, in this review, three aspects will be focused on: (1) update recent progress in the characterization of mobile sRNAs that traffic intercellular, systemic and cross-species; (2) assess functions of these mobile sRNA; and (3) discuss the possible regulatory mechanisms of sRNA intercellular trafficking in plants.

## 2. Movement of Plant Mobile Small RNAs

The mobility nature of non-cell-autonomous sRNA is supported by establishing dosage-dependent sRNA mediated developmental patterning, identifying a population of phloem mobile/grafting-transmissible sRNAs in phloem exudates and silencing of pathogen/herbivores genes by plant-derived sRNAs. In terms of the range, sRNA movement can be divided into three categories: local movement, systemic movement and cross-species/kingdom translocation.

### 2.1. Selective Local Cell-to-Cell Movement of Small RNAs through PD

Plant cells can communicate with each other through PD, a symplastic connection for intercellular exchange of signal agents, and sRNAs are among these mobile molecules. Nearly two decades ago, the siRNAs were confirmed to be non-cell-autonomous signaling agents by several groups [1,19,20]. Later, a growing number of mobile sRNAs were identified; studies of miR165/166 in maize were the first hint of miRNA mobility [21]. Then, artificial miRNAs (amiRNAs), trans-acting siRNAs (tasiRNA) and miRNAs also indicated a non-cell-autonomous effect, as shown by expressing under cell-type-specific promoters [17,22,23]. The range at which sRNAs could travel has been observed by using an artificial reporter system in which siRNAs expressed in phloem companion cells (CC) were established to be mobile siRNAs that could move 10–15 cell layers [24,25,26,27]. In limited cases, the siRNA-RISC produced mRNA aberrant RNAs are amplified to form dsRNA by RNA-dependent RNA polymerase 6 (RDR6) to produce secondary siRNAs, resulting in a wider range of chlorosis, indicating a wide range of siRNA movement [11,27]. These siRNAs are believed to move locally, cell-to-cell, through PD (Figure 1a).

Direct evidence of sRNA travel through PD came from the GFP silencing of 16c, a widely used *Nicotiana benthamiana* GFP over-expression line. Silence of GFP is induced by leaf infiltration of *Agrobacterium tumefaciens* carrying a GFP construct. The only cell type escaping from the short-range spread of RNAi-GFP signals are stomata guard cells, which are isolated from neighboring cells by occlusion of PD [24,27]. Moreover, with the larger plasmodesmata size exclusion limit (SEL), siRNA could spread through up to 35 cell layers in *Arabidopsis* embryonic tissues, compared with the 10–15 cell layers movement of RNAi signals derived from CC in leaves [28]. On the contrary, decreasing the PD permeability by increasing callose deposition, limits the non-cell-autonomous action of miR165 [29]. These observations support the notion that sRNAs move cell-to-cell through PD.

The process of intercellular movement of sRNA is directional, highly selective and cell-type dependent. The conclusion was drawn from a group of well-designed studies [17,30]. In this system, cell-type-specific promoter driven artificial miRNAs targeting GFP (amiRGFP) intercellular movement can be monitored by the loss of cell-autonomous GFP signal. In shoot apical meristem (SAM) and root apical region (RAM), amiRGFP acts cell-autonomously [17]. On the contrary, miR394 and miR390 travel intercellularly in the SAM region to maintain the stem cell niche’s function [17,30]. Furthermore, amiRGFP showed a directional movement that is distinct from the GFP protein in other cell types. Presumably, the permeability of sRNA/protein is under a highly selective process. Thus, identifying the regulators mediating sRNA cell-to-cell movement and players involved in selective processing at the PD are of great importance.

### 2.2. Mobile sRNAs Move Systemically through the Vasculature 

Long-distance signaling orchestrates different biological processes through the vasculature signal highway. Molecules detected in the phloem translation stream should be potential long-distance signal agents [5,31]. Thus, phloem mobile sRNAs should be of great potential to function as systemic signaling agents (Figure 1b). As hypothesized, systemic transmitting of RNA interference (RNAi) signal is firstly identified by grafting silenced stocks to non-silenced scions expressing *uidA* [12]. Later, the RNAi signal was identified to be the mobile signal molecules. In a parallel study, systemic silencing of GFP provides more evidence of RNAi signal systemic transmission [32]. Thus, these studies established that the mobile sRNAs are the underlying molecule function as systemic RNAi signaling agents [1,33,34,35].

In the following years, a large number of siRNAs are detected in the phloem exudates in different plant species [13,36,37,38]. Besides sequencing directly from sRNAs isolated from phloem exudates, the interspecies grafting experiments tested the population of sRNAs were moved across the graft union that is genetically distinct from the recipient tissue, allowing for determination of the origin detected mobile sRNAs [5,14,38]. Furthermore, the population of miRNAs, which can enter phloem and move to recipient tissues in response to environmental input, are going through a highly selective process [39,40]. These miRNAs might be candidates serving as information-transmitting molecules, but with grafting experiments alone, it is not sufficient to convincingly assign signaling function to the mobile sRNAs.

A number of studies showed that exogenous siRNAs introduced by inverted-repeat of gene(s), can also move intercellularly and systemically in plants [24,27,41]. The siRNA targeting *Disrupted Meiotic cDNA 1* (*DMC1*) can penetrate through grafting union to meiotically active cells, which leads to partial sterility of flowers. However, inverted GFP hairpin transcript siRNA can translocate, through grafting union, and silence GFP signals in most flower tissues, including anthers in scion [41]. Once again, suggesting in addition to a common siRNA long-distance transport pathway, a selective mechanism in the cell-to-cell penetration pathway of siRNAs.

### 2.3. Inter-Species sRNA Crosstalk

A growing body of evidence shows that sRNAs of pathogen, symbiont, parasite, and herbivores can be exchanged between host plants. These cross-species sRNAs result in degrading of target genes in recipients/invader cells, known as host-induced gene silencing (HIGS) that might function via the same pathway that siRNA intercellular, systemic movement or a new pathway as in animals (Figure 1c). Recently, a study showed siRNAs translocated through plant EVs in which host siRNAs were taken up by *Botrytis cinerea* cells to silence target genes, in a *tetraspanin* (TET)8 and TET9 dependent manner [15]. sRNAs originated from necrotrophic fungal pathogen *B. cinerea*, silence plant genes to promote its infection [42,43]. Moreover, pathogens can send effectors to mute host immunity. Among which, *Phytophthora* encoding suppressors of RNAi (PSRs), can block host secondary siRNA biogenesis and promote infection [44,45]. Once again, suggesting siRNAs as cross-kingdom signaling agents.

Signal communication, in the form of sRNAs or their precursors, between plant host and bacteria/fungus function in the establishment of the symbiont. The symbiotic bacterium Rhizobium (*Bradyrhizobium japonicum*) sends tRFs into soybean cells (*Glycine max*) to silence plant nodulation related *GmRHD3a/GmRHD3b* genes [46]. However, the mechanism of tRF translocation from Rhizobium to soybean root cells remains to be investigated. Another type of root symbiont, Arbuscular Mycorrhizal Fungus (AMF), is associated with most land plant roots. With no direct evidence of sRNA exchange reported, an in silico prediction of 237 plant transcripts as *R. irregularis* sRNAs putative targets, suggesting that AMF sRNAs should be able to translocate and silence host genes during the AMF colonization processes [47,48].

Translocation of sRNAs between host and the parasitic plant is also well established. The first evidence came from HIGS of GUS (β-glucuronidase) in parasitic plants [49]. Recently, a number of sRNAs transportation have been reported between the parasitic plant, *Cuscuta campestris*, and host plants that are involved in defense responses [16]. The parasitic plant connected with its host through vascular tissue in haustoria, which allows parasitic plants to absorb nutrients from their host. The parasitic plant connected with host through vascular tissue in haustoria, which obligate parasitic plants to absorb nutrients from the hosts. Translocation of sRNAs should occur through the phloem, which might be mediated by sRNA/RNA binding protein complexes. Identifying the agents that mediate host–parasitic plant sRNAs exchange is of great importance to reduce yield losses by parasitic plants.

## 3. Biological Function of Plant Mobile Small RNAs

### 3.1. Mobile sRNAs Regulate Plant Development 

The above-ground organs of a plant originate from the SAM cells. miRNAs, miR394 and miR165/166, are established to function as mobile signal molecules to orchestrate SAM maintenance and differentiation, likely through direct post-transcriptional regulation of key genes in the SAM (Figure 2b). In SAM cells, the miR394 spread from the site of its biogenesis, in the protoderm (L1) and the adaxial sides of the cotyledons to internal cells through PD, resulting in the degradation of the F-Box coding gene, *Leaf Curling RESponsiveness* (*LCR*). This downregulation is critical for proper WUSCHEL function in SAM [30]. Other sets of mobile miRNAs are miR165/166, which target *HD-ZIP III* transcripts *Phabulosa* and *Revoluta*. Defined cell linage of non-cell-autonomous miR165/166 mobility, by expression of miRNA locker AGO10, is essential for SAM maintenance, as SAM terminated in the *ago10* mutant display a pinhead phenotype [50]. This evidence can be an indicator that, mobile miRNAs could function as positional cues in SAM development.

Mobile miRNAs also play pivotal roles in leaf development (Figure 2c). On one hand, the miR166, expressed at the leaf abaxial side, restricts *HD-ZIPIII* to the top side, determining adaxial identity of leaf. On the other hand, tasiARF confines the *AUXIN RESPONSE FACTOR3* (*ARF3*) and *ARF4* to the bottom side of leaves, determining leaf abaxial identity [4,30,51,52,53]. Thus, the mobile sRNAs form an opposing sRNA gradient that positions developmental boundaries in leaf polar development. Another leaf development-related mobile miRNA is miR156, a grafting transmissible agent, regulates leaf morphology and trichome length on leaves and also regulates juvenile leaf identity by repressing *WUSCHEL* expression in SAM [54,55,56,57].

Mobile miRNAs are not only involved in regulating above ground organ development but also function in the underground tissues (Figure 2d). In the root, the miR165/166, originated from the endodermis cells and subsequent movement into the stellar cells, restrict the *HD-ZIP III* transcription factors expression, and thereby determine vascular patterning. Blocking the miRNA PD permeability by accumulating callose, resulting in short root phenotype similar to that of a *phb-d* or *shr* mutant. Mutual degradation of miR165/6 and *PHABULOSA* and the existence of an additional negative regulator of cytokinin signaling [51,58,59,60]. Two plasma membrane- and plasmodesmata-localized receptor-like kinases Barely Any Meristem (BAM), BAM1 and BAM2, redundantly regulate distribution of miR165/6 and their targets, for Arabidopsis root proper xylem patterning [61]. Additionally, another group of mobile miRNAs/target gene modules, miR390/tasiARF/ARF, is involved in lateral root growth [59]. Potato tuberization process is also regulated by mobile miRNAs. The miR172, a marker gene of tuberization that overexpression in aerial organs is sufficient to promote tuberization, which can transfer from *miR172* overexpression scions to wild-type stocks to regulate tuberization [62]. Another graft-transmissible mobile sRNA, miR156 can suppress potato tuberization, likely by suppressing miR172 expression through transcription factor *SPL9* (*Squamosa Promoter Binding-like 9*), to produce fewer and smaller tuber in miR156 scion grafted wild-type stocks [54].

### 3.2. Epigenetic Changes 

Beside PTGS, mobile sRNAs also function in epigenetic modifications (Figure 2a). By heterografting between mutants and TGS transgenic *Arabidopsis* plant, 24 nt sRNA have been identified as mobile signals to direct epigenetic modification, known as RNA-directed DNA methylation (RdDM) at cytosine residues, in recipient cells and RAM [11,38]. Mobile sRNAs that mediate cytosine methylation and that associate with H3K27me3, are different from non-mobile sRNAs associated loci [38,63]. For future work, characterizing functions of these mobile sRNAs is very important.

In pollen grains, sperm cell transposable elements (TEs) are suppressed by mobile sRNAs, derived from pollen vegetative nucleus TEs [64,65]. RdDM of TEs in companion cells reinforces transposon methylation in plant gametes, contributing to stable silencing of TEs across generations [66]. In the ovules, the AGO9-dependent, TEs derived mobile sRNAs from companion cells are necessary to specify epigenetic reprogramming in plant gametes [67]. Mobility of tasiR-ARF regulates female germline specification [68]). Moreover, direct evidence comes from the movement of in fluorescently labeled 24 nt sRNA injected in the central cell (CC) that is detected in egg cell (EC) [69]. After fertilization, mobile sRNAs and heterochromatic small interfering RNA (hc-siRNAs) are involved in paternal or maternal alleles imprinting [70]. Moreover, tRNAs fragment and small nucleolar RNAs (snoRNAs) can regulate transgenerational plant fitness [71]. These pieces of evidence support the model that mobile sRNAs are involved in epigenetic silencing to maintain transgenerational genome stability.

### 3.3. Abiotic Stress Responses 

Mineral nutrients are critical for plant survival, in low phosphate conditions, miR399s, induced in vascular tissues in shoot, transfer to root to silence the *UBC24* transcripts to regulate phosphate uptake [72] (Figure 2e). Recently, miR399d*, miR827, miR2111a, and miR2111a* also identified as long-distance phosphate starvation-responsive signaling molecules [73,74]. Under the stress of low-sulfur, the miR395s are induced in the root vascular system, serving as long-distance phloem-mobile signals between shoot and root [75].

Mobile sRNAs are also involved in other abiotic stresses. In drought stress conditions, a group of drought-responsive mobile miRNAs was identified. This group was expressed in the graft partner, transported between scion and rootstock to improve grapevine adaptation to drought [76]. The mobile miR172a induced by salt stress can activate *Thiamine Biosynthesis Gene* (*TTH1*) expression by cleaving *Salt Suppressed AP2 Domain-Containing* (*SSAC*) transcription factor to promote salt tolerance [77]. These mobile sRNAs are pivotal signaling agents in resulting plant systemic responses to abiotic stresses for its fitness.

### 3.4. Plant Symbiotes Interactions

A number of miRNAs, such as miR171, are critical in the regulatory network in symbiosis development [78]. Recently, an in silico analysis predicted 237 host *Medicago truncatula* putative target transcripts of fungal sRNAs [47]. Moreover, cross-species RNAi was identified, indicating the siRNAs exchange between host cells and AMF [79].

In host *Lotus japonicus* cells, upon bacterial infection, mature miR2111 is synthesized in leaves and transported to roots through the phloem (Figure 2f). The miR2111-directed cleavage of its in vivo target *Too Much Love* (*TML*), releasing the negative effect of *TML* on symbiosis formation, depending on a shoot-acting Hypernodulation Aberrant Root Formation1 (HAR1) receptor [80,81]. More recently, the rhizobial tRNA-derived small RNA fragments (tRFs) function as positive regulators of nodule symbiosis formation in soybean by targeting auxin receptors and efflux carriers, RING/U-box proteins, and protein kinases [46]. The tRFs are most likely another group of trans-kingdom mobile sRNAs signal agents.

### 3.5. Plant Pathogen Interactions

HIGS provides host plants protection against pathogens. Transgenic crop plants expressing artificial sRNAs targeting the key gene on pathogen development- and virulence-related genes, result in enhanced host resistance to pathogens, suggesting the transportation of siRNAs from host cells to pathogens [82]. Moreover, native mobile miRNAs are also identified to move cross-species targeting fungi genes. Such as cotton miR166 and miR159, that transfer to knockdown fungal *Ca^2+^-Dependent Cysteine Protease* (*Clp-1*) and *Isotrichodermin C-15 Hydroxylase* (*Hic-15*), respectively. On the other side, to establish the infection, pathogens also send sRNA effectors to host cells to facilitate their infection. For example, the *Pst-milR1* increases virulence by targeting the wheat *PR2* gene [83,84]. Fungus strains, *B. cinerea dcl1dcl2* double mutant, show reduced virulence [42,43].

### 3.6. Anti-Parasitic Plant 

Parasitic plants, such as dodders, can exchange nutrients and signaling agents through the haustoria. Host derived sRNAs transfer to parasite cells to regulate gene expression. Small RNA–mediated dodder (*Cuscuta pentagona*) *Shoot Meristemless*-like (*STM*) silencing disrupts dodder growth [85]. Besides the identified mobile exogenous- siRNA, from the host to the parasite, leading to impaired expression of essential parasite target genes. A recent study established that native host miRNA can also transfer to parasite cells [86,87]. On the other hand, sRNA can also be transported from the parasitic plant to host cells. In *Arabidopsis* and tobacco, *C. campestris* was found to deliver specific 22-nt miRNAs that are induced at the *Cuscuta* haustorium to suppress host messenger RNAs [16,88].

### 3.7. Potential Application of Mobile sRNAs in Gene Therapy

Dietary plant miRNAs are detected in mammalian blood, indicating some miRNAs are intact after storage, processing, cooking and early digestion [89,90,91,92,93,94]. It is logical to assume that dietary miRNAs could function in suppressing the complementary transcripts in recipient cells (Figure 2g). One piece of evidence is studies on miR2911 of honeysuckle (HS), a well-known Chinese herb for *Influenza A Virus* (IAV) treatment. The miR2911, as the most stable miRNA in HS, can exert a therapeutic effect on various IAVs [94]. Other dietary plant miRNAs, such as miR168a, and miR159, are detected in humans and mice blood. The miR168a can inhibit *Ldlrap1* translation. Meanwhile, plant miR159 was predominantly being inversely correlated with breast cancer incidence and progression in patients [92,95]. These plant-derived miRNAs, miR156a, miR159 and miR168a can function in another level, as they are also detected in human breast milk [96]. Plant-derived dietary sRNAs can function as cross-kingdom signaling agents, mediating a wide range of regulatory responses, pharmacological influences, and potentially provide new vistas for gene therapy.

## 4. Forms of Mobile sRNAs

The size of mobile siRNA predominately ranges between 21 to 24 nt [16,51,80]. Recently, other sized sRNAs (18 to 24 nt) are reported as a mobile regulatory signal to regulate nodule symbiosis formation in soybean [46]. Functional miRNAs and siRNAs were identified in EVs along with RNA binding proteins, including AGO1, RNA helicases (RH) 11, RH37, annexins (ANN) 1 and ANN2, indicating those proteins contributing to the mobility of sRNAs [15,97]. More recently, a new group of mobile “tiny RNAs” (10 to 17 nt) were identified inside EVs with unknown functions [98]. However, we could not exclude the possibility that sRNA movement at the pri/pre-miRNA stage, as lone noncoding RNA can serve as sRNA precursors, which could also be long-distance signaling agents.

As for whether RNAi signals spread in the ss-sRNA or dsRNA form, previously RNAi signal was believed to spread in ds-sRNA form [99]. Many Viral Suppressors of RNA-silencing (VSRs), that counteract with host RNAi by binding and blocking siRNAs movement, were identified to bind to ds-siRNAs. Such as *Tombus* virus TBSV p19, *Potyvirus TEV* HC-Pro, and *Clostero virus* BYV p21. They all bound to ds-sRNAs and blocked local and systemic RNAi to establish viral infection [100]. Furthermore, ds-sRNAs, miRNA:miRNA* RNA duplexes, were detected in the phloem translocation stream. For example, miR399* was detected in grafting transmission from miR399-over-expressing scions [101,102].

Additionally, a number of ss-sRNA binding VSRs, such as *Begomo virus* ACMV AC4 was identified [100]. Meanwhile, in the phloem exudates, a large number of ss-sRNAs were detected, while their reverse-complemented strands were not [13,36,39]. For example, miR171* was detected from phloem sap sequencing, however, the miR171 species was absent [39]. Moreover, miR827 and miR2111 were capable of long-distance movement, while their respective miRNA* species were not mobile sRNAs [101]. Lastly, native ss-RBPs, SRBP1, and PSRP1 were identified to bind and mediate cell-to-cell movement of ss-sRNA [13,103,104]. It seems sRNAs can transfer cell-to-cell in ribonucleoprotein complexes. To date, no plant ds-sRNA binding protein is identified mediating ds-sRNA intercellular and/or systemic movement.

An interesting development to note is that *Sobemovirus* RYMV P1 bound both ds/ss-sRNA and block the intercellular movement of RNAi signals [105]. In the phloem sap of *Brassica napus,* both mature miRNAs (ss-sRNA) and the miRNA* were detected [36]. Supporting that RNAi signal might be transmitted in both ds-sRNA and ss-sRNA forms.

## 5. Molecular Mechanisms of sRNA Movement

Previously, a tremendous effort had been made to identify important players involved in cell-to-cell movement of sRNAs. By using an artificial system in which siRNAs produced from CC, driven by *Sucrose Transport Protein* 2 (SUC2) promoter, targeting *Phytoene Desaturase* (PDS) or *Sulfur* (SUL) and move from CC to recipient cells, causing a visible chlorotic phenotype [25,26]. Forward genetic screens have led to the identification of the Arabidopsis *Exportin5* ortholog, *Hasty* regulates miRNA intercellular and systemic movement [5,106]. 

As discussed above, the movement of miRNA in planta is highly selective and directional [17]. By using a biochemical approach, the Phloem Small RNA Binding protein1 (CmPSRP1) identified from pumpkin (*C. maxima*) phloem, which bound selectively, and mediated the cell-to-cell movement of 24 nt sRNAs, in the form of a ribonucleoprotein (RNP) complex (Figure 3b). The stability of this RNP complex is enhanced by CmPSRPK1 phosphorylation, which permits long-distance systemic trafficking of the bound siRNA signaling agents [103]. However, CmPSRP1 homologs have yet to be identified across other plant families. More recently, a conserved Small RNA-Binding Protein 1 (SRBP1) was identified from cucurbit plants phloem exudate using RNA-affinity columns. The CsSRBP1 and its *Arabidopsis* homolog, AtSRBP1 (AtGRP7) possess an ss-sRNA binding capacity. Besides that, AtSRBP1 (AtGRP7) could travel, cell-to-cell, mediating RNAi signal intercellular movement (Figure 3a). Moreover, this protein family is conserved in different plant species, especially in important crops [104].

Another pathway of cross-kingdom sRNA movement is EVs mediated. In the extrahaustorial matrix of fungus, *G. orontii*, numerous EVs are detected of unknown origin. Indicating EVs mediate cross-species sRNA delivery (Figure 3c). The selected sRNAs, in the ribonucleoprotein complex form, might be escorted into EVs. Recently, a study showed siRNAs translocated through plant EVs in which host siRNAs were taken up by *B. cinerea* cells to silence target genes, in a *tetraspanin* (TET)8 and TET9 dependent manner [15]. EV-mediated “tiny RNAs”, from 10 to 17 nt, delivery of cross-species RNA interference to pathogen was identified [15,88,107]. However, the mechanism of cross-species sRNAs transfer between plant and parasitic plants, and the pathway of plant-derived dietary sRNAs uptake by mammalian will be required to be explored.

## 6. Conclusions

At present, we know that in higher plants, RNAs can move intercellularly, systemically to function in regulating plant responses to environmental and developmental cues. Recent studies have identified thousands of mobile RNAs through genomics approaches. Meanwhile, a well-designed experiment showed sound evidence that same amiRGFP generated different cell lineage showed different mobility, which suggested a highly selective process in sRNA in long-distance movement [17], presumably through different small RNA binding protein complexes. Identifying non-cell-autonomous players will significantly contribute to understanding the PD mediated cell-to-cell movement of siRNA by non-cell-autonomous small RNA-binding proteins [13,104]. HIGS has been established to be an important tool controlling pathogen and parasitic plant infection [15,16,86]. Recent progress established the mechanism of host siRNAs uptake by fungus pathogens through EVs. A follow-up study showed that in the EV, “tiny RNAs” (10 to 17 nt) are highly enriched [97], which are functionally unknown and have a very diverse genome origin.

In recent years, significant progress has been made in uncovering sRNA movement in a more detailed study. Nevertheless, the following fundamental mechanisms remain unknown: (1) Besides known mobile sRNA population, are there more mobile sRNAs regulating cell-type(s) specific development and responses to a/biotic stresses? (2) Besides 21 to 24 nt sRNAs, can other sizes of RNA fragments show non-cell-autonomous regulatory function? (3) Beside tRNA derived fragments, are there other types of sRNA fragments and their functions in plants? (4) What are the fundamental pathways for sRNA transmitting intercellularly, systemically or cross-species? (5) Are there more regulatory plant originated dietary sRNAs functions in regulating human health-related genes, and can they be developed as new vistas for gene therapy? Studying these open questions will hold promise for exciting findings and have broad impacts on our understanding of plant signaling and application in the near future.

## Figures and Tables

**Figure 1 plants-11-03155-f001:**
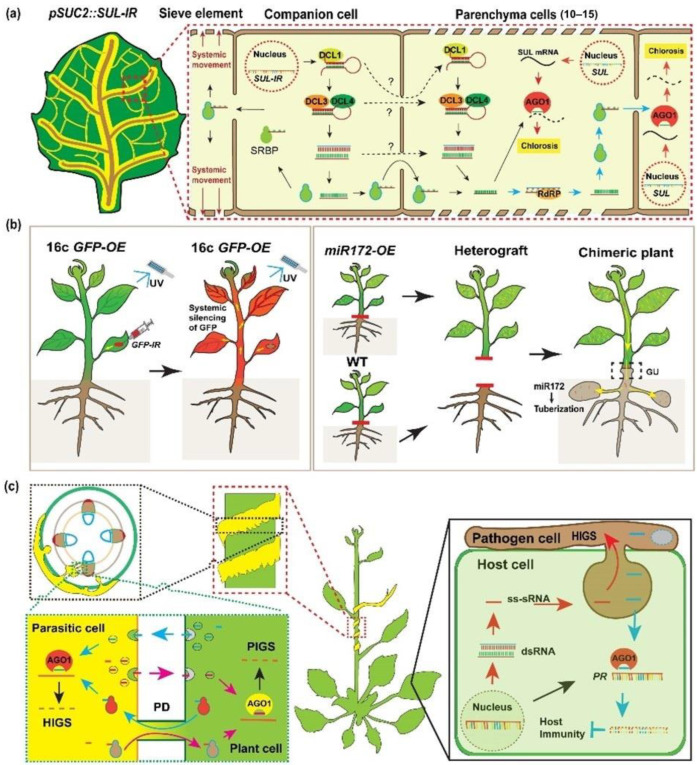
Small RNAs are mobile signals. (**a**), The RNA interfering (RNAi) signals travel locally through Plasmodesmata. The movement of sRNAs is indicated by silencing endogenous *Sulphur* (*SUL*) mRNA, which leads to chlorosis in recipient cells. The sRNAs are generated by DCLs processing inverted-repeat of *SUL* (*SUL-IR*) driven by the phloem companion cell (CC) specific gene *Sucrose transporter2* (*SUC2*) promoter, causing photobleaching phenotype in the leaves. siRNAs can move from CC to neighboring mesophyll cells through PD, up to 15 cell layers (black arrows). It is also possible that the sRNA intercellular movement might though sRNAs precursors (black dash arrows). In some cases, secondary siRNAs generated by RDR6 can move more cell layers (blue arrows). (**b**), sRNAs are long-distance signaling molecules that move through the phloem. Local introduction of GFP siRNA can trigger the systemic GFP silencing (left panel). These siRNAs are grafting transmissible through grafting union, miR172 can move from the scion, miR172 overexpression transgenic plants to stock, to trigger the tuberization of WT (right panel). (**c**), The siRNAs are cross-species signaling agents. Plant derived sRNAs can be transmitted to parasitic plants/fungal pathogens, to silencing the virulence genes, known as host induced gene silencing (HIGS); vice versa, parasitic plant/fungal pathogen sRNAs can silence host resistance genes to establish infection (PIGS).

**Figure 2 plants-11-03155-f002:**
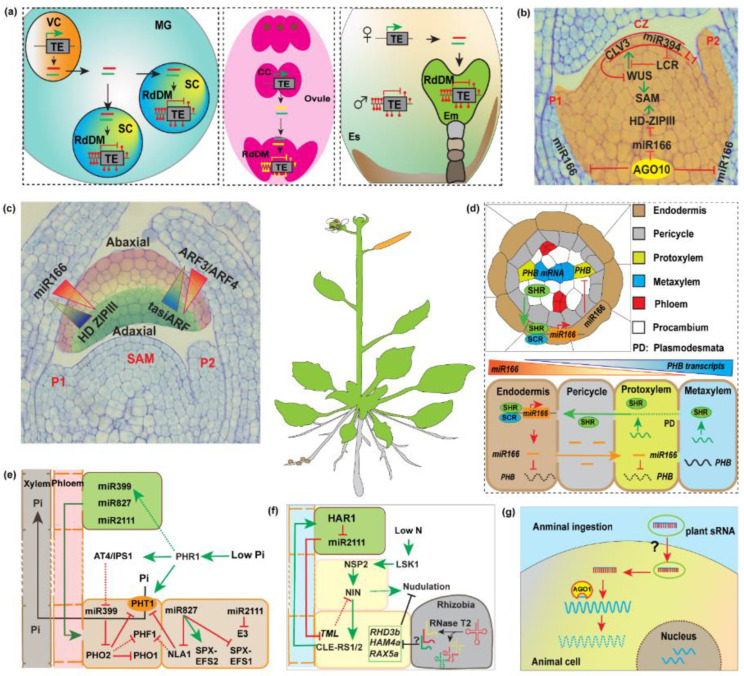
Functions of mobile sRNAs. (**a**), Mobile sRNAs regulate DNA methylation in recipient cells. In mature pollen grains (MG), the vegetative nuclei (VN) derived transposable elements (TEs) sRNAs are either targeting TE within VN or trafficking into sperm cells (SC) to silence TE by RNA-directed DNA methylation (RdDM), reinforcing TE silencing in gametes (left). In the mature embryo sac, similar to pollen grains, TE-derived siRNAs can move from the central cell (CC) to the egg cells (EC) to silence TEs (middle). During seed development, siRNAs highly produced in endosperm can move to the embryo cell and reinforce the TE silencing to stabilize the genome in the next generation (right). (**b**), Mobile miRNAs in Shoot Apical Meristem (SAM) niche maintenance. In the central zone (CZ), miR394 is expressed in the L1 cells and trafficking to L2 and L3 cells to promote *WUSCHEL* expression by suppressing *LEAF CURLING RESPONSIVENESS* (*LCR*), maintaining the WUS-CLV3 stem cell niche negative feedback loop. AGO10 sequencing miR166/165 to promote the homeodomain-leucine zipper protein (HD-ZIP) III genes (highlighted) to maintain the SAM. (**c**), Mobile miRNAs determine leaf dorsal-ventral pattern. The miR165/166 express mainly on the abaxial side and moves inward, forming a gradient accumulation of the target HD-ZIP III transcripts, which determine adaxial fate. However, tasiARFs express on the adaxial side and move to the abaxial domain, creating a targeted ARF gradient accumulation that establishes the leaf adaxial/abaxial development. (**d**), Mobile miR165/166 determine vascular development. The transcription factor SHORT ROOT moves from stele cell to endodermis to upregulate miR165/166 expression. The miR165/166 move to the inward cell layers to suppress *PHABULOSA* (*PHB*) transcripts in the protoxylem cells, which leads to the protoxylem specification. The lower levels of miR165/166 and high levels of HD-ZIP III specify the metaxylem cell identity. (**e**), Small RNAs function as long-distance signaling agents. In low phosphate conditions, root-derived Pi-stress signals are transported through the phloem to the shoot. Then, shoot-derived phloem-mobile miRNAs, such as miR399, miR827 and miR2111, travel to the root, targeting the transcripts of *PHO2*, *NLA*, and an E3 ligase, respectively, to increase the Pi uptake. (**f**), The miRNAs and tRNA fragments (tRFs) are cross-kingdom signaling agents. Shoot-derived miR2111 moves to and suppresses the *TML* in the root to trigger nodulation. Rhizobia derived tRFs can move to host cells and silence *RHA3b*, *HAM4a* and *RAX5a* to trigger nodulation. (**g**), Dietetically absorbed plants derived sRNA target genes in recipient animal cells.

**Figure 3 plants-11-03155-f003:**
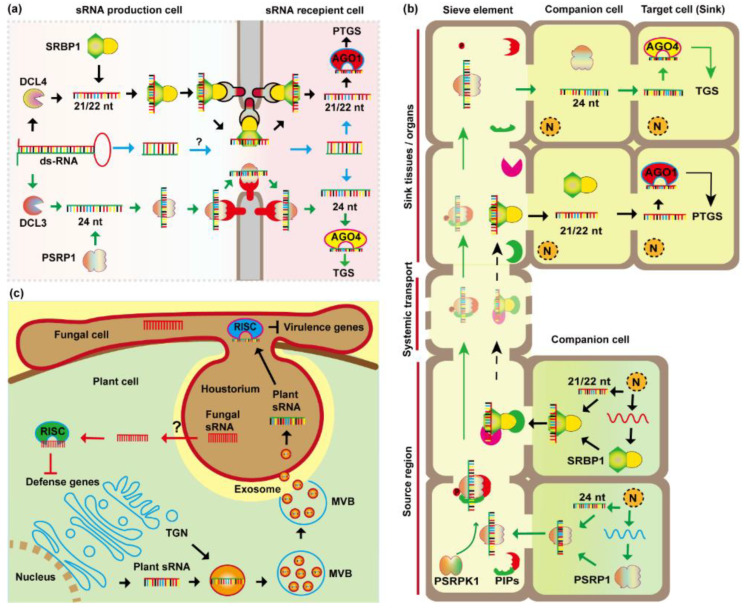
Molecular mechanism of sRNA movement. (**a**), The small-RNA binding proteins SRBP1 and PSRP1 mediate sRNA cell-to-cell movement through PD. SRBP1 binds 21/22 nt sRNA to mediate sRNAs cell-to-cell movement in SRBP1-sRNA ribonucleoprotein (RNP) form, to regulate post-transcriptional gene silencing (PTGS) through AGO1 (Black arrows). The 24 nt sRNAs bound by PSRP1 and achieve the cell-to-cell movement in RNP form to function in transcriptional gene silencing (TGS) through AGO4 (Green arrows). Another possible way is that sRNA precursors or ds-sRNA might also be able to travel intercellularly (Blue arrows). (**b**), Schematic illustrating the PSRP1/SRBP1 mediated long-distance transport of si/miRNAs. PSRP1 binds 24-nt si/miRNA in companion cells (CCs) and mediates their movement into the sieve element (SE). In the SE, CmPSRPK1 phosphorylates PSRP1, which stabilizes the PSRP1-sRNA RNP complex for long-distance transportation from source to sink tissues. In a similar approach, the 21/22 nt si/miRNAs are recognized by SRBP1 to achieve the long-distance transport to recipient tissues, to function in PTGS through AGO1. N: nucleus. (**c**), Schematic illustrating cross-species sRNA movement mediated by extracellular vesicles (EV). Fungal sRNAs travel to host cells, through yet uncharacterized mechanism(s), and hijack host RNAi-induced silencing complexes (RISC) to suppress host defense genes (Red arrows). On the other hand, plant cells send effective siRNAs into the fungal cell, through EVs, to silence fungal virulence genes (Black arrows). TGN: trans-Golgi network.

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
