# Peer review of "Insights into Mobile Small-RNAs Mediated Signaling in Plants"

_plants, 2022, doi:10.3390/plants11223155_

Round 1
Author Response
Dear Reviewer
I am greatly grateful to you for careful reading of this manuscript and helpful comments and valuable suggestions. I also want to thank you for giving suggestion on modifying our figures, which have helped me significantly to improve this manuscript. According to your suggestions, I have thoroughly revised this manuscript and its final version is enclosed. Point-by-point responses to the comments are listed below.
Reviewer 1
- In the bottom part of Parenchyma cell, what is the orange oval mark as PdRP?
Reply: It should be RdRP and corrected in the figure 1a.
- What is the difference between black arrows and black arrows with dash lines?
Reply: Black arrows indicate identified pathways in mediating mature sRNA cell-to-cell traffic, black dash arrows indicate potential transport pathways though precursors of sRNAs. It is added to line 68-69 as “The sRNA intercellular movement might also though sRNAs precursors (black dash arrows).”
- What is the green pear shape indicated?
Reply: It is the small-RNA binding proteins labeled in the figure 1a.
- What is the difference of SUL-IR and SUL?
Reply: SUL-IR stand for SUL inverted repeat (SUL-IR) sequence, it is added in the figure 1a legend in line 69-70 as “In some cases, secondary siRNAs are generated by RDR6, which moves even more cell layers (Blue arrows), SUL inverted repeat (SUL-IR).”
I suggest labeling every protein, illustrating it in the figure legend, or making the index figures
on the side to indicate different molecules such as single-strand small RNA, small RNA duplex,
and single or double-strand RNA. So far, these molecules are not clearly illustrated or
distinguished in the figure. May use the question mark to represent the unclear paths or
molecules
Reply: I am greatly appreciating your suggestions. Modification on figure 1 have been made.
Figure 3 is not mentioned in the article.
Reply: Fig 3a in cited in line 347-348 as “Besides that, AtSRBP1 (AtGRP7) could travel, cell to cell, mediating RNAi signal intercellular movement (Figure 3a).” The fig. 3b is cited in the line 339-341 as” which bound selectively, and mediated the cell-to-cell movement of 24 nt sRNAs, in the form of a ribonucleoprotein (RNP) complex (Figure 3b).” The figure 3c is cited in the line 351-353 as “In the extrahaustorial matrix of fungus, G. orontii, numerous EVs are detected of unknown origin. Indicating EVs mediate cross-species sRNA delivery (Figure 3c).”
Line 31, RNA induced silencing complexes should be RNA-induced silencing complexes
Reply: The sentence is rewritten as suggested in line 30-31 “These sRNAs can be integrated into ARGONAUTE proteins (AGOs), forming the core of RNA-induced silencing complexes (RISC),”
Line 36 Trafficking as traffic
Reply: The sequences is rewritten as “Here in this review, three aspects will be focused on: 1) update recent progress in the characterization of mobile sRNAs that traffic intercellular, systemic and cross-species; 2) assess functions of these mobile sRNA; 3) discuss the possible regulatory mechanisms of sRNA intercellular trafficking in plants.” in line 35-39.
Line 37 cross species should be cross-species; function as functions; discuss of
Reply: The sequences is rewritten as “Here in this review, three aspects will be focused on: 1) update recent progress in the characterization of mobile sRNAs that traffic intercellular, systemic and cross-species; 2) assess functions of these mobile sRNA; 3) discuss the possible regulatory mechanisms of sRNA intercellular trafficking in plants.” in line 35- 39.
Line 77 The first sentence of this paragraph is not clearly described. It would be better to
describe the 16c line and how they prove the blockage of PD is why GFP in the stomata guard
cell is not silenced.
Reply: Limited silencing movement is likely through plasmodesmata because stomata guard cells, which are simplistically isolated from neighboring cells by plasmodesmata occlusion using a Ultrastructural and histochemical method, the sentence was rewritten as “A direct evidence of sRNA travel through PD came from the GFP silencing of 16c, a widely used Nicotiana benthamiana GFP over-expression line, silencing of GFP is induced by leaf infiltration of Agrobacterium tumefaciens carrying a GFP Construct. The only cell type escaping from the short-range spread of RNAi-GFP signals are stomata guard cells, which are isolated from neighboring cells by occlusion of PD [24,27].” in line 77-81.
Line 79 What is SEL?
Reply: plasmodesmata size exclusion limit (SEL), is spelled out. “Moreover, with the larger plasmodesmata size exclusion limit (SEL), siRNA could spread through up to 35 cell layers in Arabidopsis embryonic tissues, compared with the 10-15 cell layers movement of RNAi signals derived from CC in leaves [28].” in line 82-85.
Line 129, The first time mention B.cinerea. Need to use the full name.
Reply: Full name was added in line 131-133 as “Recently, a study showed siRNAs translocated through plant EVs, in which host siRNAs were taken up by Botrytis cinerea cells to silence target genes, in a tetraspanin (TET)8 and TET9 dependent manner [15].”
Line 139, …from Rhizobium to soybean root cells remains to be invested(?). or investigated?
Reply: Sentence was rewritten as suggested in line 141-142 “However, the mechanism of tRF translocation from Rhizobium to soybean root cells remains to be investigated.”
Line 143, …AMF sRNAs should be able to translocate and “silencing” host genes during …. .
“silencing” should be “silence”
Reply: Sentence was rewritten as suggested in line 145-146 “suggesting that AMF sRNAs should be able to translocate and silence host genes during the AMF colonization processes [47,48].”
Line 158, miRNA394 should be miR394
Reply: Rewrite as suggested from miRNA394 to miR394 in line 161.
Line 214, What are BAM1 and BAM2?
Reply: BAM1 and BAM2, are receptor-like kinases. The sentence is written as “Two plasma membrane- and plasmodesmata-localized receptor-like kinases BARELY ANY MERISTEM (BAM), BAM1 and BAM2, redundantly regulate distribution of miR165/6 and their targets, for Arabidopsis root proper xylem patterning [61].” in line 216-219.
Line 223, What plant produces fewer and smaller tubers?
Reply: The miR156-OE grafted wt root stock produces less and smaller tubers. The sentence is rewritten as “Another graft-transmissible mobile sRNA, miR156 can suppresses potato tuberization, probably by suppressing the miR172 expression through transcription factor SPL9 (SQUAMOSA PROMOTER BINDING-LIKE 9), to produce fewer and smaller tuber in miR156 scion grafted wild-type stocks [54].” in line 224 -227.
Line 227, RNA directed DNA methylation, should be RNA-directed DNA methylation
Reply: Changed as suggested from “RNA directed DNA methylation” to “RNA-directed DNA methylation” in line 231.
Line 230, check the underline: In the future, characterizing functions of these mobile sRNAs
and their target loci on the molecular, morphological and phenotypic levels are very important.
Reply: Sentence is rewritten as “In the future, characterizing functions of these mobile sRNAs are very important.”
Line 239, spell out hc-siRNAs
Reply: The hc-siRNAs is spelled out as suggested and sentence was rewritten as “After fertilization, mobile sRNAs and heterochromatic small interfering RNA (hc-siRNAs) involve in paternal or maternal alleles imprinting [70].” in line 243-244.
Line 251, … a group of drought-responsive mobile miRNAs was identified, expressing expressed
in the graft partner, transporting transported between scions and rootstocks to improve
grapevine adaptation to drought
Reply: The sentence is written as “In drought stress conditions, a group of drought-responsive mobile miRNAs was identified, expressed in the graft partner, transported between scion and rootstock to improve grapevine adaptation to drought [76]. in line 255-257.
Line 255, (SSAC) should be italic.
Reply: SSAC is italic in the line 259 “salt suppressed AP2 domain-containing (SSAC)” as suggested
Line 275, What “Indicating the transportation of siRNAs from host cells to pathogens.” ?
Reply: The sentence is written as “Transgenic crop plants expressing artificial sRNAs targeting the key gene on pathogen development- and virulence-related genes, result in enhanced host resistance to pathogens, suggesting the transportation of siRNAs from host cells to pathogens [84].” In line 277-280.
Line 291, In Arabidopsis and tobacco. is not a complete sentence.
Reply: The sentence is written as “In Arabidopsis and tobacco, C. campestris was found to deliver specific 22-nt miRNAs, that are induced at the Cuscuta haustorium, to suppress host messenger RNAs [16,88].” in line294-296.
Line 294-297, No reference for the dietary plant sRNAs targeting herbivores gene. In addition,
anti-herbivores should not include in anti-parasitic plants.
Reply: It is reasonable to remove this plant sRNAs targeting herbivores gene part and Figure 2g from anti-parasitic plants section.
Line 301-302, “More recently, a new group of mobile “tiny RNAs” (10 to 17 nt) were identified
inside EVs, RNA binding proteins, AGO1, RH11 and RH37, contribute to mobile sRNAs
[15,89,90].” It is not sure if tiny RNAs contribute to mobile sRNAs. It is better to describe in
separate sentences such as: “Functional miRNAs and siRNAs were identified in EVs along with
RNA binding proteins including AGO1, RNA helicases (RH) 11, RH37, annexins (ANN) 1 and
ANN2 those contribute to sRNAs mobility [15,90]. More recently, a new group of mobile “tiny
RNAs” (10 to 17 nt) were identified inside EVs with unknown functions [89]”
Reply: Sentence is rewritten as suggested in line 300-304.
Line 307-309, spell out VSRs.
Reply: VSR is spelled out and sentence is rewritten as “As a lot of Viral Suppressors of RNA-silencing (VSRs), that counteract with host RNAi by binding and blocking siRNAs movement, were identified to bind to ds-siRNAs.” in line 308-310.
Line 319-320, “Moreover, miR827 and miR2111 were capable of long-distance movement,
while their 318 respective miRNA* species were not.” The sentence is not completed.
Reply: The sentence is rewritten as “Moreover, miR827 and miR2111 were capable of long-distance movement, while their respective miRNA* species were not mobile sRNAs [93].” in line 319-320.
Line 321, “…no plant ds-sRNA protein is…” suggested described as “ds-sRNA binding protein”
Line 349, The first mentioned G. orontii in the article. Need to use the full name.
Reply: Change “plant ds-sRNA protein” to “ds-sRNA binding protein” as suggested in line 323.
Line 355-357, “However, the mechanism of cross-species sRNAs transfer between plant and
parasitic plants, and the pathway of plant-derived dietary sRNAs uptake by mammalian are still
reported.” Dose it means been reported? Or still “not” reported, or unclear?
Reply: The molecular mechanism of cross-species sRNA movement is not clear yet. The sentence is rewritten as “However, the mechanism of cross-species sRNAs transfer between plant and parasitic plants, and the pathway of plant-derived dietary sRNAs uptake by mammalian will be required to be explored.” in line 357-359.

Reviewer 2 Report
In this manuscript, the author intended to write a review on perceptions of mobile small-RNAs mediated signaling in plants. The topic is very interesting; however, I have several concerns that need to be addressed before publication. I recommend editing the whole manuscript.
1. The manuscript is not really very well written and currently is mostly a summary.
2. The introduction section has almost nothing. The introduction section must clearly introduce the topic.
3. “Biological functions of plant mobile small RNAs” are well-known. There is no point in discussing them in detail, rather I suggest focusing more on the interesting sections such as: Inter-species sRNA crosstalk (a very important topic).
4. Section 4 (“Forms of mobile sRNAs”) and section 5 (“Molecular mechanisms of sRNA movement” must be described in more detail. The other sections in this current manuscript mostly summarize what is already known.
5. Lot of abbreviations (in fact, quite a many of them) are written without any introduction of their complete forms.
6. Please uniform the line spacing and font size.
7. Please find the attached PDF for other corrections.

Author Response
Dear Reviewer,
I would like to thank you for your insightful comments, which have helped me significantly to improve this manuscript. According to your suggestions, I have thoroughly revised this manuscript and its final version is enclosed. Point-by-point responses to the comments are listed below.
Review 2
- What about Piwi-interacting RNAs (piRNAs)? Usually, miRNAs, siRNAs and piRNAs are considered as three main classes of small RNAs.
Reply: Plants lack the third major class of small RNAs found in animals, Piwi-interacting RNAs (piRNAs), similar molecular and developmental functions have been fulfilled by siRNAs in plants. the sentence is rewritten as “These are three major groups of sRNAs in plants, small interference RNA (siRNAs), microRNAs (miRNA), and transfer RNA-derived fragments (tRFs), which are 21 to 24 nucleotide (nt) in size [1-4].”in line 21-23
- Please mention the full-name here.
Reply: Editing has been made as suggested and rewritten as “The siRNAs are generated through processing double-stranded RNAs (dsRNAs), either inverted-repeat (IR) transcripts or RNA-dependent RNA polymerases (RDRs) synthesized dsRNAs, by Dicer-Like2 (DCL2), DCL3 or DCL4 [5]. The miRNAs are generated by DCL1 cleavage of imperfectly paired harpin of native transcript precursors [6].” in line 24-27.
- Please change to "Dicer-Like1 (DCL1)"
Reply: Editing has been made as suggested and rewritten as “The siRNAs are generated through processing double-stranded RNAs (dsRNAs), either inverted-repeat (IR) transcripts or RNA-dependent RNA polymerases (RDRs) synthesized dsRNAs, by Dicer-Like2 (DCL2), DCL3 or DCL4 [5]. The miRNAs are generated by DCL1 cleavage of imperfectly paired harpin of native transcript precursors [6].” in line 24-27.
- There are no mention of PIGs in the whole text
Reply: PIGS is inserted in the end of sentence in line 76 “The siRNAs are cross-species signaling agents. Plant derived sRNAs can be transmitted to parasitic plants/fungal pathogens, to silencing the virulence genes, known as host induced gene silencing (HIGS); vis versa, parasitic plant/fungal pathogen sRNAs can silencing host resistance gene, to establish infection (PIGS).”
- What is transacting RNA? You mean trans-acting RNA?
Reply: Yes, it should be trans-acting RNA, the sentence is written as “Later, artificial miRNAs (amiRNAs), trans-acting siRNAs (tasiRNA) and miRNAs also showing a non-cell-autonomous effect as showed by expressing under cell-type-specific promoters [17,22,23].” in line 52-54
- please make it uniform? Previously you wrote dsRNA, without hyphen.
Reply: Many thanks for pointing it out, the sentence is written as “In limited cases, siRNA:targets dsRNAs can be amplified by RNA-dependent RNA polymerase 6 (RDR6) to produce secondary siRNAs, resulting in a wider range of siRNAs cell-to-cell movement [11,27].” in line 57-59 as suggested.
- SEL No full form mentioned anywhere.
Reply: Full name is inserted in the sentence as suggested “Moreover, with the larger plasmodesmata size exclusion limit (SEL), siRNA could spread through up to 35 cell layers in Arabidopsis embryonic tissues, compared with the 10-15 cell layers movement of RNAi signals derived from CC in leaves [28].” in line 82-84.
- amiRGFP, What is the full form? There are no mention of it in the upper part.
Reply: Full name is inserted in the sentence as suggested “The conclusion was drawn from a group of well-designed studies. In this system, cell-type-specific promoter driven artificial miRNAs targeting GFP (amiRGFP) intercellular movement can be monitored by the loss of cell-autonomous GFP signal.” in line 89-91.
- Please mention the full form: Sterile alpha motif (or SAM)
Reply: SAM in this case stand for shoot apical meristem. Full name is inserted in the sentence as suggested “On the contrary, miR394 and miR390 travel intercellularly in the shoot apical meristem (SAM) region, to maintain stem cell niches function [17,30].” in line 92-94.
- Plant symbiotes interactions:??
Reply: Thank you for pointing out this mistake, the “:” is removed in the revised version in line 276.

Round 2
Reviewer 2 Report
Please find my comments in the attached file. The manuscript is improved but still I feel that moderate English editing is required. The figure qualities are really very bad. Please improve them before resubmission.
